

# Neither heat pulse, nor multigenerational exposure to a modest increase in water temperature, alters the susceptibility of Guadeloupean *Biomphalaria glabrata* to *Schistosoma mansoni* infection

Euan R.O. Allan[1], Stephanie Bollmann[2], Ekaterina Peremyslova[2] and Michael Blouin[2]

[1] School of Veterinary Medicine, Pathobiology, St. George's University, St. George's, Grenada
[2] Integrative Biology, Oregon State University, Corvallis, OR, United States of America

## ABSTRACT

There are increasing concerns regarding the role global climate change will have on many vector-borne diseases. Both mathematical models and laboratory experiments suggest that schistosomiasis risk may change as a result of the effects of increasing temperatures on the planorbid snails that host schistosomes. Heat pulse/heat shock of the BS90 strain of *Biomphalaria glabrata* was shown to increase the rate of infection by *Schistosoma mansoni*, but the result was not replicable in a follow up experiment by a different lab. We characterised the susceptibility and cercarial shedding of Guadeloupean *B. glabrata* after infection with *S. mansoni* under two temperature regimes: multigenerational exposure to small increases in temperature, and extreme heat pulse events. Neither long-term, multigenerational rearing at elevated temperatures, nor transient heat pulse modified the susceptibility of Guadeloupean *B. glabrata* to infection (prevalence) or shedding of schistosome cercaria (intensity of infection). These findings suggest that heat pulse-induced susceptibility in snail hosts may be dependent on the strain of the snail and/or schistosome, or on some as-yet unidentified environmental co-factor.

## INTRODUCTION

Parasitic trematodes can cause numerous mammalian diseases. The most notable and detrimental human disease, schistosomiasis, is caused by the blood fluke *Schistosoma mansoni*. Schistosomiasis is estimated to affect hundreds of millions of people a year, exert massive health and economic tolls on tropical countries, and contribute to hundreds of thousands of deaths (*WHO/Department of Control of Neglected Tropical Diseases, 2012*; *WHO/Department of Control of Neglected Tropical Diseases, 2016*). Freshwater snails of the genus *Biomphalaria* serve as obligate intermediate hosts for *S. mansoni* (Sm). *Biomphalaria glabrata* (Bg) is the most important host for *S. mansoni* in the new world. Schistosome eggs from infected human host feces release miracidia, which penetrate and infect snails.

Corresponding author
Euan R.O. Allan, eallan1@sgu.edu, euanroallan@gmail.com

The parasites transform and develop in these snails over a few weeks to become cercariae, which are shed from the snail and go on to infect human hosts and cause disease.

It has been postulated that increasing average temperatures, as a result of global climate change, will affect the distribution, population sizes, fecundity, survival, and transmission dynamics of many species involved in vector-borne diseases, including *Biomphalaria sp.* (*McCreesh et al., 2014*; *McCreesh & Booth, 2014*; *McCreesh, Nikulin & Booth, 2015*). In the context of snail infection, both parasite and host are fully exposed to external abiotic factors. Miracidia are free swimming and actively infect ectothermic snails in aquatic ecosystems. As such, the physiology and immunity of Bg has been shown to be modified by transient changes in water temperature (*Ittiprasert & Knight, 2012*; *Knight et al., 2015*; *Knight et al., 2016*; *Nelson et al., 2016*; *Sullivan, 2018*; *Coelho & Bezerra, 2006*; *Augusto, Duval & Grunau, 2019*). Models of long-term increases in water temperate in natural ecosystems suggest that *Biomphalaria* populations and infection dynamics may lead to increases in human and snail infection risk (*McCreesh & Booth, 2014*). Concurrently, heat wave exposure is postulated to increase as a result of global climate change as natural fluctuations in temperature are exacerbated (*Peng et al., 2011*). Heat waves can result in spikes in water temperature and heat shock of ectotherms.

Heat shock of the highly resistant BS90 strain of Bg was shown to increase the susceptibility of this strain to schistosome infection, and the effect appeared to be mediated by expression of heat shock proteins (*Ittiprasert & Knight, 2012*; *Knight et al., 2015*; *Knight et al., 2016*). On the other hand, a follow up study in a different lab failed to replicate the increase in susceptibility in BS90 after heat pulse, a result which suggests that there may be specific effects of the strain of BS90 or Sm used, or that some unidentified environmental co-factor is involved (*Nelson et al., 2016*; *Sullivan, 2018*).

Models of the effects of climate change on schistosomiasis risk could be improved by understanding any changes in transmission rate that result from temperature change *per se*. Therefore, it seems important to determine how generalizable the effect of heat pulse is in other populations of Bg and Sm. We set out to determine if a long-term subtle increase in temperature or a transient heat pulse (both ecologically relevant with global climate change), can alter the infection dynamics of another snail-schistosome pair: Guadeloupean Bg (BgGUA) challenged with Guadeloupean Sm (SmGUA).

To examine the effects of a modest but consistent increase in ambient temperature, we maintained BgGUA at 1 °C above their standard conditions for 7 months (>3 generations at 27 °C vs the standard 25–26 °C). Though there are huge variations in the estimate of increases to the average temperature in tropical freshwater aquatic ecosystems over the past few decades, we believe that 1 °C (27 °C) is a reasonable conservative estimate (*McCreesh et al., 2014*; *Pekel et al., 2016*; *Engels et al., 2019*). Additionally, mathematical models suggest that *Biophalaria* populations may begin to crash when in natural aquatic ecosystems above a consistent water temperature of 28 °C (*McCreesh & Booth, 2014*). In a separate experiment, we transiently heat pulsed (32 °C) BgGUA for 6 h before challenge to determine if their susceptibility to SmGUA was altered by heat pulse (and likely heat shock) responses. Given the heterogeneity of heat pulse infection phenotypes and adaptability of schistosomes to numerous geographical ecosystems, we hypothesised that infection of BgGUA with

SmGUA would not be altered by changes in water temperature. We found that neither long-term multigenerational rearing at elevated temperatures nor transient heat pulse modified BgGUA's susceptibly to challenge by SmGUA. These findings suggest the effects of heat on susceptibility of Bg to Sm may be constrained to specific circumstances.

## MATERIALS AND METHODS

### *Biomphalaria glabrata* maintenance and ethics

Snails (BgGUA) were collected from Guadeloupe in 2005 and maintained as previously described unless otherwise stated (*Theron et al., 2008*; *Theron et al., 2014*; *Allan & Blouin, 2017*; *Allan et al., 2017a*; *Allan et al., 2017b*; *Allan & Blouin, 2018*; *Allan et al., 2019*; *Allan et al., 2018*). All snails were kept in dechlorinated water. Heat pulses conducted using incubators, with constant water temperature monitoring, and long-term small temperature modifications were conducted using thermal stratification in a single room with hourly followed by daily water temperature monitoring. All experiments followed the Public Health Service Domestic Assurance for humane care and use of laboratory animals (PHS Animal Welfare Assurance Number A3229-01), as Animal Care and Use Proposal 4360; approved by Oregon State University Institutional Animal Care and Use Committee.

### BgGUA long-term temperature exposure and infection studies

To examine the effects of a modest but long-term temperature increases, BgGUA were maintained in 25 °C (standard for some other strains e.g., BS-90), 26 °C (standard for BgGUA), or 27 °C (elevated) for >7 months (all temperatures monitored daily). Adult snails (>12 mm) were allowed to mate, lay eggs, and be in the presence of the juveniles until the juveniles reached >3 mm; after which all juveniles were re-tanked (thus eliminating the inclusion of egg sacs from previous generations) and the previous generation was sacrificed. This was done 3 times to ensure >3 generations were maintained at a specific temperature. Schistosome challenges were carried out as previously described with some modifications (*Ittiprasert & Knight, 2012*; *Sullivan, 2018*; *Allan et al., 2017a*; *Allan et al., 2017b*). In brief, size matched (∼7 mm) BgGUA were individually challenged with 10 miracidia for 12 h in 2 ml of dechlorinated water, and transferred into tubs containing up to 10 snails each ($n = 41, 48, 46$ for 25 °C, 26 °C, or 27 °C respectively). All challenges, and maintenance post-challenge, was conducted in water corresponding to the experimental maintenance temperature (25 °C, 26 °C, or 27 °C). Weekly (from week 5 to week 10), snails were placed under light for 3 h in 24 well dishes in 2 ml of dechlorinated water (at 25 °C, 26 °C, or 27 °C) and examined for cercarial shedding. They were scored as infected or uninfected, the number of cercaria shed per infected snail was counted, and non-shedding snails were returned to the tank for future assessment ($n = 18, 18, 21$ for 25 °C, 26 °C, or 27 °C respectively). All proportions are cumulative for the 10 week period, and cercarial counts are from the first shedding incidence before infected snails were sacrificed.

### BgGUA heat pulse and infection studies

To assess the effects of heat pulse, size matched (∼7 mm) BgGUA were removed from standard conditions (26 °C) and exposed to 26 °C (control), or 32 °C (heat pulse) for 6 h

prior to challenge with miracidia. After removal from temperature exposure, snails were immediately challenged with 10 miracidia at 26 °C to ensure no effects of temperature on schistosome activity. Challenges were carried out as previously described (*Allan et al., 2017a*; *Allan et al., 2017b*). BgGUA were individually challenged for 12 h in 2 ml of dechlorinated water, transferred into tubs containing up to 10 snails each, and monitored for 10 weeks. Weekly (from week 5 to week 10), snails were placed under light for 3 h in 24 well dishes in 2 ml of dechlorinated water and examined for cercarial shedding, scored as infected or uninfected ($n = 48$ and 49 for 26 °C, or 32 °C respectively), and the number of cercaria shed per snail was counted ($n = 15$ for both treatments), and non-shedding snails were returned to the tank for future assessment. All proportions are cumulative for the 10 week period, and cercarial counts are from the first shedding incidence before infected snails were sacrificed.

## Statistical analyses

Statistical analyses on the number of cercaria shed were completed by one-way ANOVA (or unpaired Student's *t*-test) with a Tukey-test, while analysis of the proportion infected was done by calculating the $Z$ score (standard score) of the population ($p < 0.05$) (*Allan et al., 2017b*). Analyses were completed using GraphPad Prism software (La Jolla, CA, USA).

## RESULTS

### BgGUA maintained at 1 °C above standard conditions for multiple generations, or transiently heat pulsed 6 °C above standard conditions, do not show altered susceptibility to SmGUA

Snail susceptibility to infection, as a proportion, was recorded as a measure of the ability of a given snail to resist infection, and has relevance for the number of disease transmitting snail hosts. Snail burden of infection, measured by cercarial count during first shedding (only shedding event quantified), was recorded as a measure of transmission risk by each snail. BgGUA, regardless of maintenance temperature (25 °C, 26 °C, or 27 °C), had equivalent susceptibility and cercarial shedding when exposed to SmGUA (Figs. 1A and 1B). Additionally, heat pulse (32 °C) did not alter the susceptibility or the number of cercaria shed by BgGUA (Figs. 2A and 2B). Most snails shed cercaria by week 6–7 regardless of treatment (Data S1). No snail mortality was observed, though infected snails were sacrificed after shedding and could have feasibly died from the infection if they were returned to the population rather than sacrificed.

## DISCUSSION

As the potential impacts of global climate change on vector-borne diseases become more evident, and climate change worsens, the importance of understanding these ramifications are accentuated. Our findings support the hypothesis that neither permanent nor transient increases in temperature alter BgGUA susceptibility to infection by SmGUA. Though we are the first to examine prolonged temperature increases in BgGUA, it has been previously discussed that subtle changes in temperature do not generally alter infection dynamics in some other snail-schistosome combinations (*Ittiprasert & Knight, 2012*). Given the

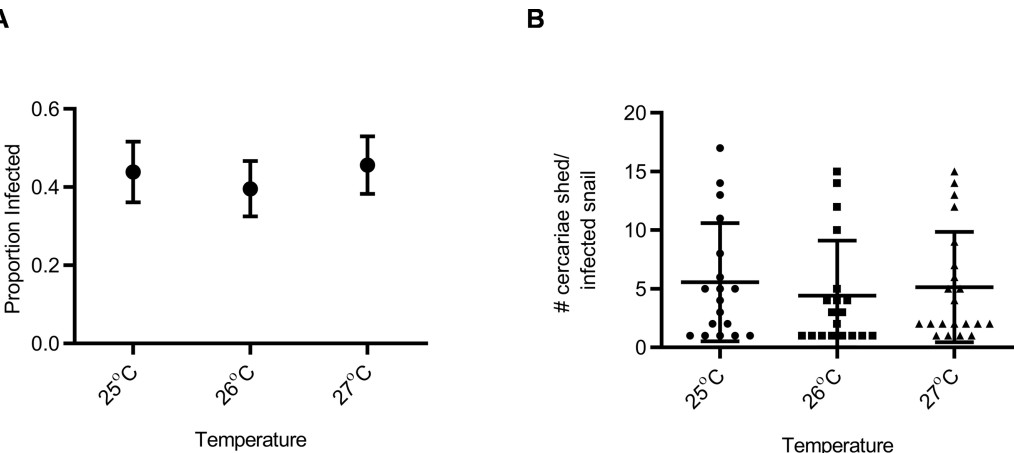

**Figure 1 Susceptibility and cercarial shedding of BgGUA is not altered by multigenerational mainte-nance at elevated temperature.** (A) Susceptibility of BgGUA maintained for >7 months at 25 °C, 26 °C, or 27 °C. Data are presented as proportion of infected snails +/− the standard error of proportions ($n$ = 41, 48, 46 for 25 °C, 26 °C, 27 °C). (B) The total number of cercariae released over 3 h single shedding event by infected snails. Data are presented as mean +/− SD ($n$ = 18, 18, 21 for 25 °C, 26 °C, 27 °C). No significant differences (Z score of proportion; ANOVA $p > 0.05$).

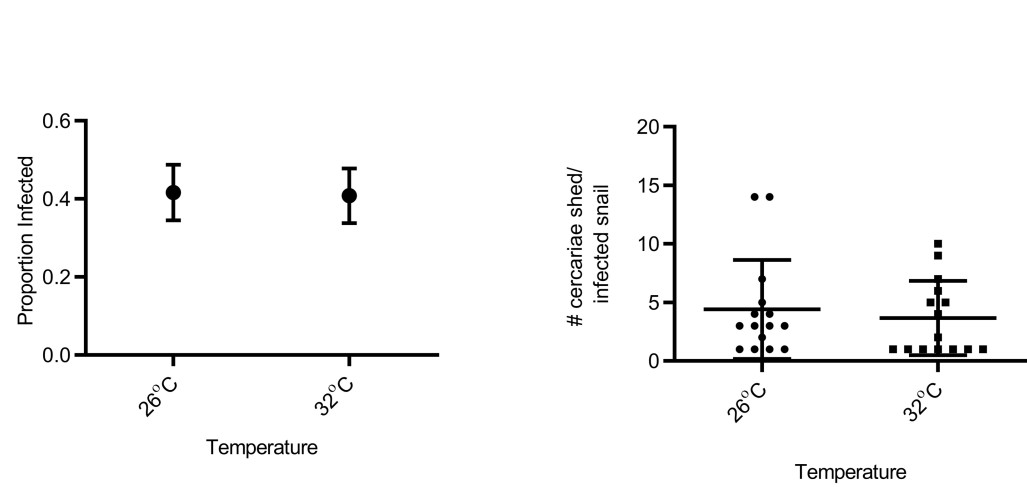

**Figure 2 Susceptibility and cercarial shedding of BgGUA is not altered by 6 h heat pulse prior to schis-tosome challenge.** (A) Susceptibility of BgGUA pulsed for 6 h at 26 °C or 32 °C (heat pulsed) immediately preceding schistosome challenge. Data are presented as proportion of infected snails +/− the standard er-ror of proportions ($n$ = 48, 49 for 26 °C, 32 °C). (B) The total number of cercariae released over 3 h single shedding event by infected snails. Data are presented as mean +/− SD ($n$ = 15). No significant differences (Z score of proportion; Student $t$-test $p > 0.05$).

strain specific nature of schistosome infectivity, it was important to determine if these long-term temperature changes could modify this strain's susceptibility. Our findings confirm that long-term/subtle elevations in temperature, as a result of climate change, may not modify snail resistance to schistosome challenge. They also speak to the robustness

of snail-schistosome infectivity dynamics. Additionally, the burden of SmGUA infection, measured by counting cercarial shedding, was equivalent under all temperature conditions, and therefore temperature fluctuations are not likely to alter schistosome transmission risk from individual BgGUA. These findings also support the notion that the effects of heat pulse/heat shock on schistosome resistance are not general, and are probably strain specific or dependent on particular environmental conditions (*Nelson et al., 2016*; *Sullivan, 2018*).

Specifically, it is important to note that BgGUA strain is less resistant to schistosomes than the strain used in other heat pulse experiments, the BS90 strain. Therefore, our study provides insights into the effects of heat treatments on more susceptible strains of snails, which we believe weakens the conclusion that heat waves could increase infectivity in nature. It is feasible that snails which are already susceptible to schistosome challenge would be even more vulnerable to abiotic effects increasing susceptibility, but we do not observe this. We also believe the conclusion that heat shock increases snail susceptibility could be further confounded by the possible effects of abiotic components like water quality, humidity, diet, and infection conditions. These subtle variations in the environment could disrupt the effects of temperature, thus weakening or masking the overall importance of heat pulse on schistosome infectivity.

## CONCLUSIONS

Although it does not appear that an increase in susceptibility to Sm following heat pulse is a general result, it is important to remember that temperature and climate can affect many biotic factors (including fecundity, growth rates, mortality, and mobility) that influence the transmission of Sm. Controlled laboratory infections are not necessarily representative of what will occur in nature. For example, it is possible that a natural population of Bg could become more susceptible to Sm, but that they have reduced fecundity resulting in the transmission of fewer parasites (*McCreesh et al., 2014*). Permanent climatic changes could also shift the ranges of these species because they can only tolerate a finite increase in temperature (*McCreesh & Booth, 2014*; *McCreesh, Nikulin & Booth, 2015*). These interacting factors could create different, but not necessarily larger, regions of high schistosomiasis risk. As such, it is important to determine how geographically distinct Bg and Sm respond to changes in abiotic factors. Determining the direct effects of these changing environmental conditions on schistosome infection and risk, in a local context, will allow for more accurate schistosomiasis risk models.

### Funding
This work was supported by the National Institutes of Health (AI109134). The funders had no role in study design, data collection and analysis, decision to publish, or preparation of the manuscript.

### Grant Disclosures
The following grant information was disclosed by the authors:
National Institutes of Health: AI109134.

## Competing Interests

The authors declare there are no competing interests.

## Author Contributions

- Euan R.O. Allan conceived and designed the experiments, performed the experiments, analyzed the data, prepared figures and/or tables, authored or reviewed drafts of the paper, and approved the final draft.
- Stephanie Bollmann and Ekaterina Peremyslova performed the experiments, prepared figures and/or tables, and approved the final draft.
- Michael Blouin conceived and designed the experiments, authored or reviewed drafts of the paper, funding, and approved the final draft.

## Data Availability

Raw data is available as Supplemental File.

## Supplemental Information

Supplemental information for this article can be found online at http://dx.doi.org/10.7717/peerj.9059#supplemental-information.

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
