# Peer review of "Neither heat pulse, nor multigenerational exposure to a modest increase in water temperature, alters the susceptibility of Guadeloupean Biomphalaria glabrata to Schistosoma mansoni infection"

_PeerJ, doi:10.7717/peerj.9059_

## Round 0.1 · original submission · Major Revisions

The review process is now complete, and three thorough reviews from highly qualified referees are included at the bottom of this letter. All reviewers including myself agree the manuscript deserves to be published. Although there is considerable merit in your paper, we also identified some concerns that must be considered in your resubmission. The Results Section deserves to be improved and completed as recommended by the reviewers. Please, also consider addressing the Discussion Section accordingly to reviewers’ comments and suggestions.

Reviewer 1 ·

Basic reporting

This manuscript is well written, and the rationale for the study is justified. The relevant literature is cited, and sufficient background is given. The only thing missing is the hypothesis that is being tested. A suggested hypothesis, i.e., that weak resistance is more easily reversed than strong resistance, is explained below.

Experimental design

This paper is timely in that it addresses a discrepancy in the literature, i.e., the failure of one laboratory to repeat the result of another that found that heat shock, and specifically upregulation of the Hsp90 gene in response to a brief increase in ambient temperature, induced susceptibility to infection in a schistosome-resistant snail. This question is of obvious importance in schistosomiasis epidemiology, especially with rising global temperatures. The experiments are well designed and the results are unambiguous.

Previous studies on effects of elevated temperature on resistance used the BS90 snail, which is highly resistant to infection. It would be informative to state in the Introduction what the susceptibility phenotype of the BgGUA snail is to infection with SmGUA. Based on the approximately 40% infection prevalence in control snails, it appears partly nonsusceptible, and indeed this would be a reasonable justification or hypothesis for this study, i.e., to see if weak resistance is more susceptible to heat treatment than the near complete resistance of BS90 snails.

Since these results are in effect a refutation of those of the original study claiming reversal of susceptibility by heat treatment, minor experimental details that differ should be noted. For example, were snails incubated in distilled water overnight prior to exposure to high temperature, and were the high temperatures obtained with the use of a water bath, as in the Ittaprasert and Knight study?

On line 111, the term "age matched" is used, but it seems as though "size matched" may be more accurate, since snails of the same size can be of markedly different ages.

Validity of the findings

The results are unambiguous and the interpretation straightforward. As mentioned above, that even a low degree of resistance or nonsusceptibility (versus that in BS90 snails) is not reversible by heat treatment is a novel twist that further weakens the case for the claim of susceptibility reversal, and it perhaps bears mentioning.

The use of "heat shock" in the title and text should be considered. My understanding of heat shock is exposure to an elevated temperature that elicits the heat shock response, which is characterized by expression of heat shock proteins (Hsps). Ittiprasert and Knight and Nelson et al. documented that their treatments caused upregulation of Hsp70 and Hsp90 in BS90 snails, i.e., heat shock, and the former authors showed with the use of an Hsp90 inhibitor that it was this specific protein that was responsible for susceptibility reversal. Although it is reasonable to assume that similar treatments would cause heat shock in another strain of Biomphalaria glabrata, without evidence of elevated expression of Hsp genes, can you claim that the snails in this study were heat shocked?

Additional comments

No further comments.

Reviewer 2 ·

Basic reporting

No comment

Experimental design

Most of my remarks relate to the material and methods used for this study. Please see general comments for the authors.

Validity of the findings

Supplementary experiments could be done to reinforce the conclusions. Please see general comments for the authors.

Additional comments

The paper entitled “Neither heat shock, nor multigenerational exposure to a modest increase in water temperature, alters the susceptibility of Guadeloupean Biomphalaria glabrata to Schistosoma mansoni infection.” presents information about susceptibility of the snail Biomphalaria, the major intermediate host for Schistosoma mansoni, in a context of global warming. Here, sympatric snail population from Guadeloupe were infected with a compatible parasite strain for which a prevalence around 50% is expected. The topic is interesting especially that previous conflicting studies carried out two groups on the same snail strain (Bs90, qualified as resistant) lead to different conclusions. Abstract and Introduction are well written. The results are clear and well discussed.
Most of my remarks relate to the material and methods used for this study.

Cercarial shedding was determined weekly during 10 weeks. It would have been more informative if the counting has been proceeded as described in Theron et al, 1997 Experimental parasitology. According to me, cumulative cercarial production per snail is the only relevant criterion to conclude on the increased risk (or not) of transmission from infected snails.
Also, no observation about the length of the prepatent period was given. Please discuss about this point.
Have the parental population, the first and second generation, been successively removed from the tank? No detail is mentioned while the authors indicated studying the third (at least) generation.


Possible Suggestions.
Line 54 : please cite for review Augusto RC et al, 2019 Frontiers Microbiol.

In previous study, the authors selected susceptible and resistant snail lines from Guadeloupean population. It could be relevant to demonstrate if the results are identical from different snail genotype and/or phenotype (as suggested by the authors in the abstract).

A snail heat shock was performed at 32°C for 6h. Did the authors observe an increase transcription of two molecular markers, HSP70 and 90 as described by Knight and coll. ? In my mind, it would be judicious to monitor their expression to discuss their relevance.


Addressing these points may require minor revisions and supplementary experiments, but will significantly improve the quality of the manuscript, which has great potential to provide novel insight into Biomphalaria - Schistosoma complex interaction in the face of global warming.

Reviewer 3 ·

Basic reporting

The manuscript by Allan et al is a succinct report that suggests that temperature-based modification of snail susceptibility to Schistosoma may not be widespread. The manuscript is clearly presented and touches on appropriate references; particularly those by Knight et al, and Sullivan that deal with the same topic.

Additional clarity could be provided in the results section, which is incredibly brief. The title of the singe results section is almost as long as the text. It would seem to me that a bit more detail could be provided here.

In addition, there is a statement made in the discussion, line 168, that states that 'particular environmental conditions' may impact snail susceptibility. A bit more detail could also be provided here with respect to what the conditions might be.

Experimental design

The study design is relative simple and easy to follow. It is unclear, however, whether the cumulative snail infection prevalence and cercarial shedding amount is being reported, or whether only the 10-week time point is being reported. The authors state in the methods that the snails were monitored for a period of 10 weeks, but no time series is presented.

If the snails were in fact monitored weekly for 10 weeks, it may be worthwhile for the authors to present the cumulative data alongside the data on a per-week basis. Perhaps the impact of temperature in this infection system is observed in the time to cercarial shedding, or the amount of cercariae shed early in the infection.

The authors do not address snail mortality in the methods or results. Since snails could have died over the course of the study, particularly those that were placed in the heat shock treatment, it would comforting to know whether mortality rates in each treatment group differed and how that was addressed when calculating infection rate.

Validity of the findings

This study incorporates a reasonable number of snails for each treatment. Again, I'd like to know more about the monitoring process, as this could expand the scope of the manuscript slightly to include a temporal aspect to the analysis.

Additional comments

In the introduction, it is stated that multigenerational rearing at each temperature was undertaken for this study (line 84). This is not really discussed in the manuscript, and seems as though it is an important aspect of the study from the perspective of more accurately representing a natural population. Moreover, it is one aspect of this study that has not been undertaken by other temperature-based susceptibility studies, at least to my knowledge. More emphasis could be placed on this aspect of the study in the discussion, and should be explained in the methods.

---

## Round 0.2 · Minor Revisions

Your revised manuscript still deserves attention. Please, correct the text according to the comments made the Reviewer #1 in the new version of your manuscript.

Reviewer 1 ·

Basic reporting

No additional comments. The authors made the suggested changes.

Experimental design

Fine. Additional information on temperature exposure was provided.

Validity of the findings

These results confirm 2 other papers that failed to replicate an earler study showing the effect of elevated temperature on susceptibility to infection. The solid experimental design, sample sizes, analysis of results, and discussion establish the validity of the paper.

Additional comments

“Schistosoma” is mis-spelled in the title.
Line 30. Possessive form of a species name could be misleading, especially since "'s" is italicized. I suggest “susceptibility of Guadeloupian B. glabrata” at the end of the sentence.
Line 53. An abbreviation (Sm) is previously given for the parasite on line 44, but not for the snail.
Line 168. Should “[6]” be “[9]? A cursory reading of [6] does not reveal any study of subtle temperature changes (although I may have missed it, since the paper is quite dense), whereas [9] reports on susceptibility of adult BS-90 snails held at 20 or 33 C for 1 or 2 weeks prior to challenge. If [9], then 1-2 week or short term would be more accurate than subtle, since these are relatively extreme temperatures.
Line 171. Change many to may.
Just to close the circle, the authors may wish to add in the Discussion the statement that "the results support our hypothesis that..."

Reviewer 2 ·

Basic reporting

no comment

Experimental design

no comment

Validity of the findings

no comment

Additional comments

All my remarks have been taken into account.So, i have no additionnal comment.

Reviewer 3 ·

Basic reporting

n/a

Experimental design

n/a

Validity of the findings

n/a

Additional comments

The authors have addressed my initial concerns.

---

## Round 0.3 · accepted · Accept

All points raised by the reviewer were well addressed.